# A Benchmark for Compositional Visual Reasoning

**Aimen Zerroug**[1,2,3], **Mohit Vaishnav**[1,2,3], **Julien Colin**[2], **Sebastian Musslick**[2], **Thomas Serre**[1,2]

[1] Artificial and Natural Intelligence Toulouse Institute, Université de Toulouse, France
[2] Carney Institute for Brain Science, Dept. of Cognitive Linguistic & Psychological Sciences
Brown University, Providence, RI 02912
[3] Centre de Recherche Cerveau et Cognition, CNRS, Université de Toulouse, France
{aimen_zerroug, mohit_vaishnav, julien_colin,
sebastian_musslick, thomas_serre}@brown.edu

## Abstract

A fundamental component of human vision is our ability to parse complex visual scenes and judge the relations between their constituent objects. AI benchmarks for visual reasoning have driven rapid progress in recent years with state-of-the-art systems now reaching human accuracy on some of these benchmarks. Yet, there remains a major gap between humans and AI systems in terms of the sample efficiency with which they learn new visual reasoning tasks. Humans' remarkable efficiency at learning has been at least partially attributed to their ability to harness compositionality – allowing them to efficiently take advantage of previously gained knowledge when learning new tasks. Here, we introduce a novel visual reasoning benchmark, Compositional Visual Relations (CVR), to drive progress towards the development of more data-efficient learning algorithms. We take inspiration from fluid intelligence and non-verbal reasoning tests and describe a novel method for creating compositions of abstract rules and generating image datasets corresponding to these rules at scale. Our proposed benchmark includes measures of sample efficiency, generalization, compositionality, and transfer across task rules. We systematically evaluate modern neural architectures and find that convolutional architectures surpass transformer-based architectures across all performance measures in most data regimes. However, all computational models are much less data efficient than humans, even after learning informative visual representations using self-supervision. Overall, we hope our challenge will spur interest in developing neural architectures that can learn to harness compositionality for more efficient learning.

## 1 Introduction

Visual reasoning is a complex ability requiring a high level of abstraction over high dimensional sensory input. It highlights human's capacity to manipulate concepts and relations as symbols extracted from visual input. The efficiency with which humans learn new visual concepts and relations, as exemplified by fluid intelligence and non-verbal reasoning tests, is equally fascinating. In the pursuit of human-level artificial intelligence, a growing body of research is attempting to emulate this skill in machines, and deep neural networks are at the forefront of the field.

Deep learning approaches are prime candidates as models of human intelligence due to their success at learning from data while relying on simple design principles. However, these architectures are imperfect models of human intelligence, as shown by their lack of sample efficiency, the inability to generalize to unfamiliar situations [13] and the lack of robustness [14]. Their ability to perform well

36th Conference on Neural Information Processing Systems (NeurIPS 2022) Track on Datasets and Benchmarks.

in large-data regimes has skewed research towards scaling up datasets and architectures with little consideration for the sample efficiency of these systems.

Only a few benchmarks address these aspects of human intelligence. One such benchmark, ARC [9] provides diverse visual reasoning problems. However, the extreme scarcity of training samples, only 3 samples per task, renders the benchmark difficult for all methods, especially neural networks. Other benchmarks have led to the development of new neural network-based models that address particular gaps between human and machine intelligence [3, 43, 12]. Some focus on evaluating the task's perceptual requirements [12], which include detecting features, recognizing objects, perceptual grouping and spatial reasoning. Others evaluate logical reasoning requirements [3, 43], such as symbolic reasoning, making analogies and causal reasoning. However, they lack either the variety of abstract relations present in the scene or the semantic and structural variety of scenes over which they instantiate these abstract relations.

Creating novel visual reasoning tasks can be challenging. In this benchmark, we standardize a process for creating tasks compositionally based on an elementary set of relations and abstractions. This process allows us to exploit a wide range of visual relations as well as abstract rules, thus, making it possible to evaluate both the perceptual and logical requirements of visual reasoning. The compositional nature of the tasks provides an opportunity to investigate the learning strategies wielded by existing methods. Among these methods, we focus on state-of-the-art abstract visual reasoning models and standard vision models. These models have been

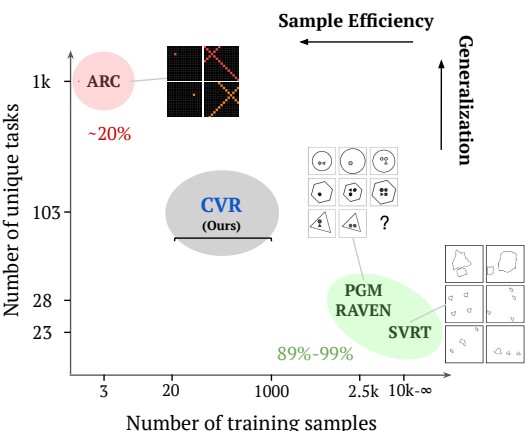

Figure 1: **Visual reasoning benchmarks**: State-of-the-art models achieve super-human accuracy [40, 38] on several visual-reasoning benchmarks such as RAVEN [43] PGM [3] and SVRT [12]. However, some benchmarks continue to pose a challenge for current models, such as ARC [9]. The fundamental difference between these different benchmarks is the number of unique task rules they composed out of their priors and the number of samples available for training architectures on individual rules. This difference sheds light on two poorly researched aspects of human intelligence: learning in low-sample regimes and harnessing compositionality. The proposed CVR challenge aims to fill the gap between current benchmarks to encourage the development of more sample-efficient and more versatile neural architectures for visual reasoning.

shown to reach high performance on several visual reasoning tasks in previous works [40, 38], but they always require large amounts of data. This paper's subject of interest is quantifying these models' sample efficiency.

**Contributions**  Our contributions can be summarized as follows:

- A novel visual reasoning benchmark called **Compositional Visual Relations** (CVR) with 103 unique tasks over distinct scene structures.
- A novel method for generating visual reasoning problems with a compositionality prior.
- A systematic analysis of the sample efficiency of baseline visual reasoning architectures.
- An empirical study of models' capacity at using compositionality to solve complex problems.

Our large-scale experiments capture a multitude of setups, including multi-task and individual task training, pre-training with self-supervision on dataset images to contrast learning of visual representations vs. abstract visual reasoning rules, training over a range of data regimes, and testing transfer learning between dataset tasks. We present an in-depth analysis of task difficulty, which provides insights into the strengths and weaknesses of current models. Overall, we find that the best baselines trained in the most favorable conditions fall short of human sample efficiency for learning those same tasks. While models appear to be capable of transferring knowledge across tasks, we show that they do not leverage compositionality to efficiently learn task components. We hope to inspire research on more efficient visual reasoning models by releasing our dataset. The code for generating the full dataset and training models is available here.

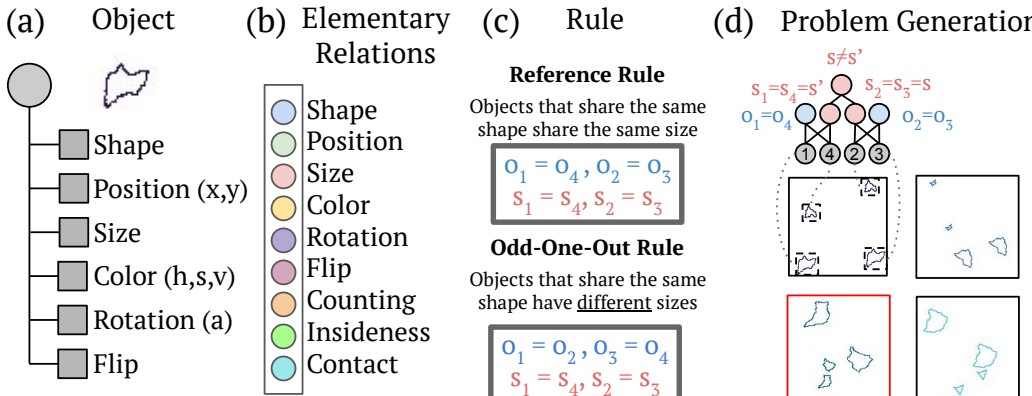

(a) Object
(b) Elementary Relations
(c) Rule
(d) Problem Generation

Figure 2: **Scene Generation**: A scene in our image dataset is composed of objects. (a) An object is a closed contour with several attributes. (b) A relation is a constraint for the generation process over scene attributes. (c) The elementary relations control unique scene attributes. They are used for building task rules in a compositional manner. Each task uses a Reference rule and an Odd-One-Out rule to generate images. (d) Odd-One-Out problems are randomly generated using a program. Three images are generated following the Reference rule, and a fourth image (highlighted in red) is generated following the Odd-One-Out rule.

## 2 Compositional Visual Relations Dataset

CVR is a synthetic visual reasoning dataset that builds on prior AI benchmarks [12, 9] and is inspired by a cognitive science literature [37] on visual reasoning. In the following, we will describe the generation process of the dataset.

**Odd-One-Out**   The odd one out task has been employed in prior work to test visual reasoning [27]. A sample problem consists of 4 images generated such that one of them is an outlier according to a rule. The goal of the task is to select the outlier. The learner is expected to test several hypotheses in order to detect the outlier. This process requires them to infer the hidden scene structure and relationships between the objects.

**Scene generation**   Each image contains one **scene** composed of multiple **objects** as shown in Figure 2. An object is defined as a closed contour with a set of **object attributes**: *shape*, *position*, *size*, *color*, *rotation* and *flip*. Other attributes describe the scene or low-level relations between objects. *Count* corresponds to the number of objects, groups of objects or relations. *Insideness* indicates that an object contains another object within its contour. *Contact* indicates that two object contours are touching. These 9 attributes are the basis for the 9 **elementary relations**. For example, a "size" relation is a constraint on the sizes of certain objects in the scene. Relations are expressed with natural language or logical, relational and arithmetic operators over scene attributes. Relations and objects are represented as nodes in the **scene graph**. Relations define groups of objects and can have attributes of their own. Thus, it is possible to create abstract relations over these relations' attributes. A scene can be generated from a template that we call a **structure**. The concepts of structure, scene graph and relations are used for formalizing the process behind designing a task. In practice, the

---

**Algorithm 1: Problem Generation Program**: Generates problem samples of the shape-size task in Figure 2

$n \leftarrow 4$                 // Number of objects
**for** $i \leftarrow 1$ **to** 4 **do**
  $s \leftarrow sample\_size()$
  $s' \leftarrow s \times rand([2/3, 1/4])$
  **if** $i = 4$ **then**
    // Odd-One-Out
    $[s_i]^{1-n} \leftarrow [s, s', s, s']$
  **else**
    $[s_i]^{1-n} \leftarrow [s, s, s', s']$
  **end**
  $[o, o'] \leftarrow sample\_shapes(n = 2)$
  $[o_i]^{1-n} \leftarrow [o, o, o', o']$
  $[p_i]^{1-n} \leftarrow sample\_position([s_i]^{1-n})$
  $[c_i]^{1-n} \leftarrow sample\_color(n = 1)$
**end**
$[scene]_{1-4} = [[o, p, s, c]^{1-n}]_{1-4}$
$[image]_{1-4} = [render(scene)]_{1-4}$

---

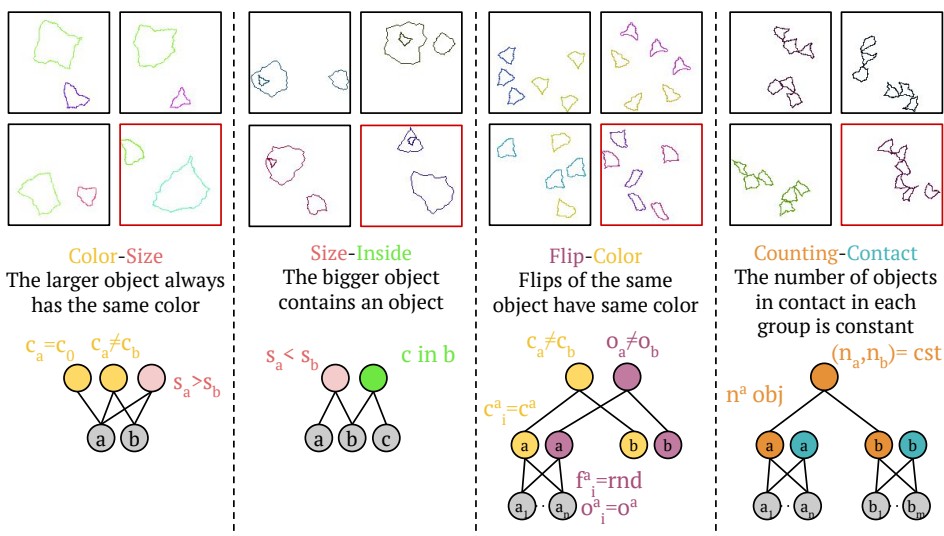

Figure 4: **Examples of task rules that are composed of a pair of relations.** More examples of tasks and algorithms are provided in the SI.

**generation process** is a program implemented by the task designer to generate problem samples of one task randomly. The Pseudo-code for an example program is detailed in Alg. 1.

**Rules and problem creation**   The generation process described above can be used to instantiate different tasks; binary classification, few-shot binary classification, or a raven's progressive matrix. In this paper, we choose to apply this process to create odd-one-out problems. First, the task designer selects target relations and incorporates them into a new scene structure. In Figure 2, the target relations are size and shape similarity; they are added to a scene with 4 objects. Then, a reference rule and an odd rule are chosen such that they combine target relations in different ways. The reference and odd rules in the example vary only in the size or shape attributes. A valid odd-one-out rule contradicts the reference rule such that any strategy used to solve the task must involve exclusively reasoning over the target relations. Given a scene structure, a reference and an odd-one-out rule, the generation process has a set of free parameters that control the generation process for new samples. The problem's difficulty level can be varied by randomizing or fixing these parameters. In the shape-size task, the range of color values and the variation of objects across the 4 images are examples of free parameters. More random parameters result in a higher difficulty. We create generalization test sets by changing the sets of fixed or random parameters. For more details on the generalization test sets we refer the reader to the SI.

**Dataset details**   CVR incorporates 103 unique reference rules, including 9 rules instantiating the 9 elementary visual relations and 94 additional rules built on compositions of the relations.

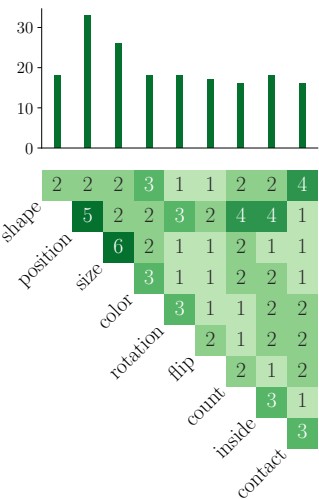

Figure 3: **Dataset rules**: Each square represents the number of rules that are a composition of the associated elementary relations and the bar plot shows the number of rules that involve each elementary relation.

These compositions span all pairs of elementary rules and include up to 4 relations. While some rules are composed of the same elementary relations, they remain unique in their scene structure or associations with other relations. 20 are compositions of single elementary relations, 65 are compositions of a pair of relations and 9 are compositions of more than 2 elementary relations. Figure 3 details the number of unique rules for each pair of elementary relations. The procedural generation of problem samples helps us create an arbitrary number of samples. We create 10,000

| | | 20 | | 50 | | 100 | | 200 | | 500 | | 1000 | | SES | AUC | 10000 | |
|---|---|---|---|---|---|---|---|---|---|---|---|---|---|---|---|---|---|
| **N train samples** | | | | | | | | | | | | | | | | | |
| rand-init / ind | ResNet-50[15] | 28.0 | 1 | 31.1 | 1 | 32.5 | 3 | 34.0 | 6 | 38.7 | 12 | 44.8 | 24 | 33.7 | 34.9 | - | - |
| | ViT-small[11] | 28.6 | 1 | 30.1 | 4 | 30.9 | 4 | 31.9 | 4 | 33.8 | 4 | 35.1 | 7 | 31.3 | 31.7 | - | - |
| | SCL[40] | 26.9 | 0 | 30.0 | 1 | 30.3 | 2 | 30.0 | 2 | 31.4 | 2 | 33.4 | 5 | 29.9 | 30.3 | - | - |
| | WReN[3] | 30.0 | 0 | 32.0 | 2 | 32.9 | 2 | 34.1 | 3 | 36.3 | 6 | 39.0 | 15 | 33.4 | 34.1 | - | - |
| | SCL-ResNet 18 | **31.4** | 1 | **37.3** | 9 | **37.8** | 9 | **39.6** | 15 | **42.7** | 21 | **48.3** | 26 | **38.4** | **39.5** | - | - |
| rand-init / joint | ResNet-50 | **27.5** | 0 | 28.2 | 0 | 29.9 | 2 | 33.9 | 6 | **52.1** | 29 | 59.2 | 34 | 36.0 | 38.4 | **93.7** | **93** |
| | ViT-small | 27.3 | 1 | 27.8 | 2 | 28.0 | 1 | 28.1 | 1 | 29.9 | 2 | 31.4 | 3 | 28.4 | 28.7 | 58.7 | 37 |
| | SCL | 25.8 | 0 | 25.8 | 0 | 28.3 | 1 | 34.1 | 3 | 43.2 | 22 | 46.2 | 27 | 32.2 | 33.9 | 56.9 | 34 |
| | WReN | 26.8 | 0 | 27.6 | 0 | 28.5 | 0 | 30.1 | 0 | 36.4 | 9 | 42.3 | 20 | 30.9 | 32.0 | 64.5 | 43 |
| | SCL-ResNet 18 | 26.4 | 0 | **28.4** | 0 | **31.6** | 4 | **40.7** | 13 | 51.4 | 32 | **64.0** | 42 | **37.6** | **40.4** | 78.9 | 73 |
| SSL / ind | ResNet-50 | 40.5 | 13 | 47.3 | 18 | 52.9 | 29 | 56.8 | 34 | 61.9 | 42 | **67.7** | 50 | 52.4 | 54.5 | - | - |
| | ViT-small | **46.7** | 16 | **51.6** | 24 | **54.8** | 29 | **57.5** | 38 | **62.0** | 44 | 65.5 | 46 | **54.9** | **56.4** | - | - |
| SSL / joint | ResNet-50 | **44.3** | 16 | **50.3** | 24 | **55.3** | 30 | **59.5** | 42 | **68.9** | 49 | **79.2** | 59 | **57.0** | **59.6** | 93.1 | 97 |
| | ViT-small | 39.3 | 15 | 39.5 | 13 | 40.8 | 14 | 44.1 | 16 | 53.3 | 30 | 60.7 | 41 | 44.7 | 46.3 | 81.6 | 67 |
| IN / joint | ResNet-50 | **32.0** | 2 | **35.1** | 5 | **39.0** | 9 | **43.8** | 13 | **57.7** | 48 | **69.5** | 48 | **43.4** | **46.2** | - | - |
| | ViT-small | 27.9 | 2 | 28.2 | 1 | 28.6 | 2 | 30.0 | 2 | 35.6 | 5 | 47.2 | 24 | 31.7 | 32.9 | - | - |
| CLIP / joint | ResNet-50 | 28.7 | 0 | 32.0 | 2 | 40.8 | 11 | 46.9 | 18 | 59.7 | 40 | 74.4 | 53 | 43.7 | 47.1 | - | - |
| | ViT-base | **31.1** | 1 | **37.4** | 7 | **43.9** | 14 | **56.0** | 30 | **68.9** | 48 | **78.8** | 62 | **48.9** | **52.7** | - | - |

Table 1: **Performance comparison**: For each model, we report the accuracy and number of tasks with accuracy above 80%. SES is the Sample Efficiency Score; it favors models with high performance in low data regimes and consistent accuracy across regimes. SES and AUC are computed over the 20-1000 data regimes. OOD generalization results are provided in the SI.

training problem samples, 500 validation samples and 1,000 test samples for each task. We also create a generalization test set of 1000 samples.

We define the compositionality prior as the task's design constraint which ensures that solving the task requires reasoning over its elementary components. In the size-shape task, shown in figure 2, the outlier can be differentiated from the other images by reasoning purely on size and shape. In the context of CVR, compositional extends beyond combinations of object attributes, such as novel color and shape combinations in an object, to higher levels of abstractions; groups of objects and scene configurations. For example, the position-rotation composition rule in Fig. 4 requires reasoning over the rotation properties of two sets of objects in each scene, and the position properties of objects within each set.

CVR constitutes a significant extension to the Synthetic Visual Reasoning Test (SVRT) [12] in that it provides a systematic reorganization based on an explicit compositionality prior. Among the 23 SVRT tasks, many share relations, such as tasks #1 and #21, which both involve shape similarity judgments. Most of these tasks can still be found amongst CVR's rules. At the same time, CVR is more general because it substitutes binary classification tasks with odd-one-out tasks which allows one to explore more general versions of these tasks, with a broader set of task parameters. For example, in SVRT's task #7, images of 3 groups of 2 same shapes are discriminated from images of 2 groups of 3 same shapes. This task is a special case in CVR of a more general *shape-count* rule with $n$ groups of $m$ objects where the values are randomly sampled across problem samples. Unlike procedurally generated RPM benchmarks [3, 43], CVR does not rely on a small set of fixed templates for the creation of task rules. The shapes are randomly created and positions are not fixed on a grid (for most rules), which renders the visual tasks difficult for models that rely on rote memorization [20]. Other attributes are sampled uniformly from a continuous interval.

## 3 Experimental setting

**Baseline models** In our experiments, we select two vision models commonly used in computer vision. We evaluate ResNet [15], a convolutional architecture used as a baseline in several bench-

marks [3, 43, 38] and also used as a backbone in standard VQA models. We also evaluate ViT, a transformer-based architecture [11]. ViT is used for various vision tasks, such as image classification, object recognition, captioning and recently in visual reasoning on SVRT [28]. To compare the architectures fairly, we choose ResNet-50 and ViT-small, which have an equal number of parameters. Additionally, we evaluate two baseline visual reasoning models designed for solving RPMs: SCL [40] which boasts state-of-the-art accuracy on RAVEN and PGM, and WReN [3] which is based on a relational reasoning model [33]. Finally, we present SCL-ResNet-18 which consists of an SCL with ResNet as a visual backbone thus combining ResNet's perception skills with SCL's reasoning skills.

**Joint vs. individual rule learning**   Models are either trained in a single task (individual) or multi-task (joint) setting. In the context of the multi-task training on CVR, one image is considered an odd-one-out with respect to a reference rule. However, because of the randomness of scene generation, a different image might be considered an odd-one-out with respect to a different, irrelevant rule. To illustrate this problem, let's take the elementary size rule as an example. In this rule, each image contains one object. Due to the random sampling of object attributes, it is possible for one image to be considered an outlier with respect to the color rule (The attributes in the 4 images are i-small/green, ii-large/green, iii-small/green, iv-small/blue). Without specifying that the task to solve involves a size relation, the model could incorrectly choose the fourth image because it is an outlier with respect to the color rule. Thus, models trained on several tasks could easily confound rules. To avoid this problem, models are provided with a rule embedding vector. Given the rule token, models can learn several strategies and use the correct one for each problem sample. We also compare the multi-task and single task settings, as they allow for testing the model's capacity and efficiency at learning several strategies and routines to solve different rules. All hyperparameter choices and training details are provided in the SI.

**Self-Supervised pre-training**   Unlike humans who spend a lifetime analyzing visual information, randomly initialized neural networks have no visual experience. To provide a more fair comparison between humans and neural networks, we pre-train baseline models on a subset of the training data. Self-Supervised Learning (SSL) has seen a rise in popularity due to its usefulness in pre-training models on unlabelled data. By using SSL, we aim to dissociate feature learning from abstract visual reasoning in standard vision models. We pre-trained ViT-small and ResNet-50 on 1 million images from the dataset following MoCo-v3 [8]. In addition to SSL pre-trained models, we also finetune models pre-trained on object recognition and image annotation. Since image annotation requires visual reasoning capabilities, these pretrained models provide a more fair comparison with humans, who regularly perform the task. We select ResNet-50 and ViT-small pre-trained on ImageNet [10]. We also pick CLIP [31] visual encoders ResNet-50 and ViT-Base, which are trained jointly with a language model on image annotation.

| N training samples | 20 | | 1000 | |
| --- | --- | --- | --- | --- |
| ResNet-50 | 28.0 | 0 | 57.9 | 14 |
| ViT-small | 29.3 | 1 | 32.7 | 3 |
| SCL | 26.4 | 0 | 44.9 | 11 |
| WReN | 27.5 | 0 | 42.4 | 10 |
| SCL-ResNet 18 | 26.8 | 0 | **64.1** | **18** |
| ResNet-50 SSL | 45.7 | 7 | **78.3** | **25** |
| ViT-small SSL | 38.7 | 6 | 60.3 | 17 |
| Humans | **78.7** | **26** | - | - |

Table 2: **Human Baseline**: performance of models on joint training experiments is compared to the human baseline. The analysis is restricted to the 45 tasks used for evaluating humans. ResNet 50 approaches human-level performance only after SSL pre-training and finetuning on all task rules with 1000 samples per rule. Which is 50 times higher than the number of samples needed by humans.

**Human Baseline**   As found in [12], having 21 participants solve the 9 tasks based on elementary relations and 36 randomly sampled complex tasks is sufficient to yield a reliable human baseline. We used 20 problem samples for each task which corresponds to the lowest number of samples used for training baseline models. Each participant completed 6 different tasks. More details about the behavioral experiment are provided in the SI.

## 4   Results

**Sample Efficiency**   Baseline models are trained in six data regimes ranging from 20 to 1000 training samples. All sample efficiency results are summarized in Table 1. Randomly guessing yields 25%

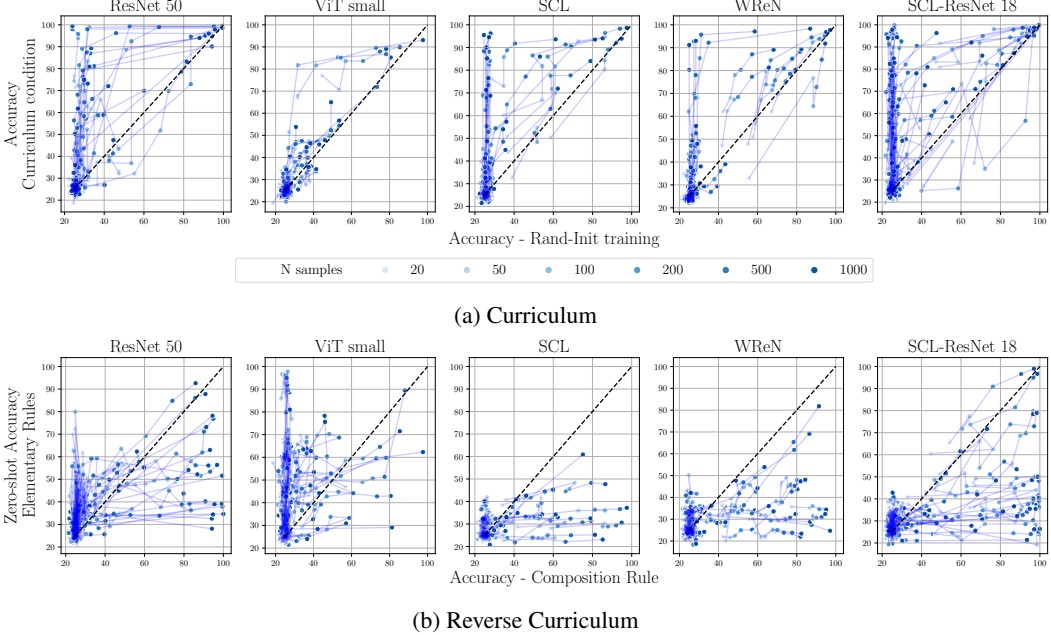

(a) Curriculum

(b) Reverse Curriculum

Figure 5: **Compositionality**: We evaluate models' capacity to reuse knowledge. (a) Models trained with a curriculum are compared to models trained from scratch. Models trained with a curriculum are overall more sample efficient. (b) Models trained on compositions are evaluated zero-shot on the respective elementary rules. Models fail overall to generalize from compositions to elementary rules.

accuracy. We observe that most randomly initialized models are slightly above chance accuracy after training in low data regimes. They achieve an increase in performance only when provided with more than 500 training samples. SCL-ResNet-18 performs the best in high data regimes, followed by ResNet-50. SCL and ViT have the lowest performance in high data regimes. This result is unsurprising since transformer architectures generally learn better in high data regimes (millions of data points). This is consistent with prior work [38] which finds that ViTs do not learn several SVRT tasks even when trained on 100k samples. Although SCL's performance is near chance, it achieves the best performance when it is augmented with a ResNet-18, which is a strong vision backbone. This jump in performance is indicative of the two architectures' complementary roles in visual reasoning. Results in Table 1 and Fig. 6 show a clear positive effect of pretraining on all models. SSL pre-trained models achieve the highest performance compared to object recognition and image annotation pretrained models. We observe that ViT benefits from a larger architecture coupled with pre-training on a large image annotation dataset. This highlights transformers' reliance on large model sizes and datasets.

In order to quantify sample efficiency systematically for all models, we compute the area under the curve (AUC), which corresponds to the unweighted average performance across data regimes. We also introduce the *Sample Efficiency Score* (SES) as an empirical evaluation metric for our experimental setting. It consists of a weighted average of accuracy where the weights are reversely proportional to number of samples: $SES = \frac{\sum_n a_n w_n}{\sum_n w_n}$ where $w_n = \frac{1}{1+\log(n)}$ and $n$ is the number of samples. This score favors models that learn with the fewest samples while considering consistency in the overall performance. We observe that SCL-ResNet-18 scores the highest in the individual and joint training settings. In the SSL finetuning condition, ViT and ResNet-50 have a similar SES when trained on individual tasks, but ResNet-50 performs better in the joint training setting. These results hint at the efficiency of convolutional architectures in visual reasoning tasks. Collapsing across all data regimes and training paradigms, the best performance on CVR is given by ResNet-50, in the joint training setting with 10k data points per rule. It achieves 93.7% accuracy. This high performance in the 10,000 data regime demonstrates the models' capacity to learn the majority of rules in the dataset and suggests that failure in lower data regimes is explained by their sample inefficiency.

Finally, we compare model performance to the human baseline. We observe in Table 2 that humans far exceed the accuracy of all models with only 20 samples. This result aligns with previous work on the SVRT dataset [12] where participants solved similar tasks with less than 20 samples. These results highlight the gap between humans and machines in sample efficiency and emphasize the need to develop more sample-efficient architectures.

**Compositionality** Transferring knowledge and skills across tasks is a crucial feature of intelligent systems. With our experimental setup, this can be characterized in several ways. A compositional model should reuse acquired skills to learn efficiently. Thus, when it is trained on all rules jointly, it should be more sample efficient because the rules in the dataset share elementary components. In Table1 and Figure6, we observe that ResNet-50 achieves higher performance on joint training compared to individual rule training, while ViT has the opposite effect. The trend is consistent across data regimes and other settings. These results highlight convolutional architectures' learning efficiency compared to transformer architectures.

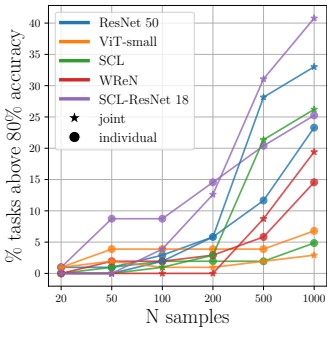

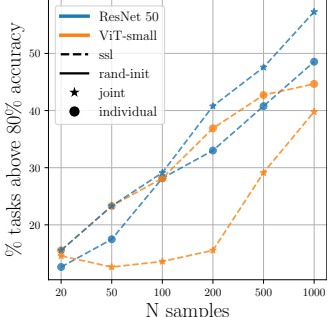

We investigate compositionality further by asking whether learning elementary rules provides a good initialization for learning their compositions. For example, a model that can judge object positions and sizes should not require many training samples to associate sizes with positions. We pick a set of complex rules with at least two different elementary relations, train models to reach the maximum accuracy possible on component relations, then finetune the models on the compositions. We call this experimental condition the curriculum condition since the condition is akin to incrementally teaching routines to a model. We compare model performance in the curriculum condition to performance when training from scratch. The results highlighted in Figure 5a show positive effects for most models but more significantly for convolution-based architectures. These results indicate that the baselines use skills acquired during pre-training to learn the composition rules, and that this pretraining helps to varying degrees. We refer the readers to the SI for additional analyses and quantitative results.

Figure 6: **Sample efficiency**: The percentage of tasks for which performance is above 80% plotted against the number of training samples per task rule, with random initialization (top) and with SSL pre-training (bottom).

Finally, we evaluate transfer learning from composition rules to elementary rules. We name this condition the reverse curriculum condition. The working hypothesis is that models that rely on compositionality will be able to solve elementary relations without finetuning if they learn the composition. We compare performance on a composition rule to zero-shot accuracy on the respective elementary rules in Figure 5b. We observe that all models perform worse on the elementary relations. These results might indicate that although the baselines could transfer skills from elementary rules to their compositions, they do not necessarily use an efficient strategy that decomposes tasks into their elementary components. Additional analyses are presented in the SI.

**Task difficulty** We analyze the performance of all models in the standard setting: joint training on all rules from random initialization. Figure 7 shows the average performance of each model on each elementary rule and composition rule. Since the dataset contains several compositions of each pair of elementary rules, the accuracy shown in each square is averaged over composition rules that share the same pair of elementary rules. Certain rules are solvable by all models, such as the *position*, *size*, *color*, and *count* elementary rules. Additionally, other rules pose a challenge for all models, these rules are compositions of *count*, *flip*, *rotation* or *shape*. Models that rely on a convolutional backbone were able to solve most spatial rules; *position*, *size*, *inside* and *contact*. However, they fail on rules that incorporate shapes and their transformations; *shape*, *rotation*, *flip*. Composition rules built with the *Count* relation proved to be a challenge for most models. We believe that models are capable of solving several tasks, such as the *counting* elementary rule, by relying on shortcuts; this could be a summation of all pixels in the image, for example. These shortcuts prevent models from learning

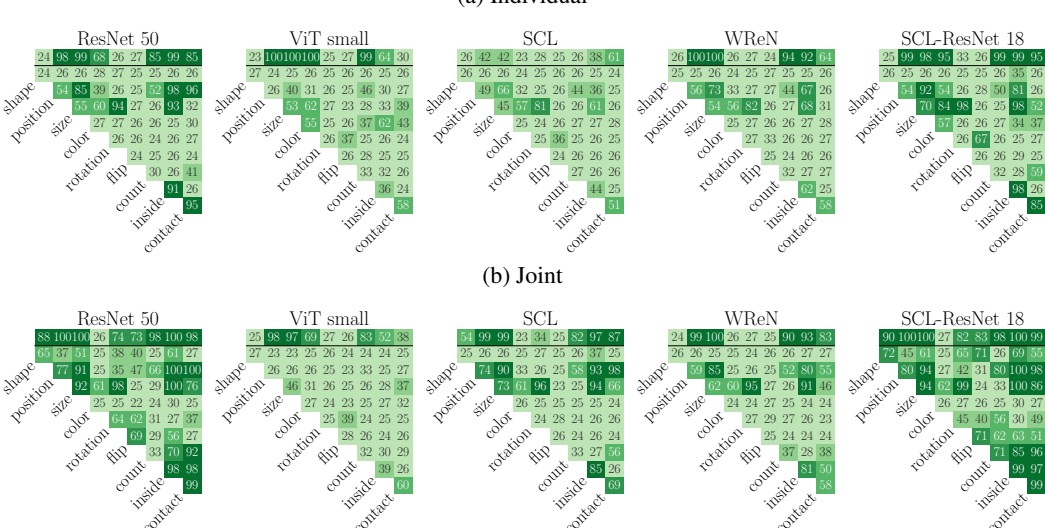

Figure 7: **Task analysis**: The performance at 1000 samples is shown for each model. Performance on elementary rules is shown on the top row of each matrix. The elementary relations of each composition are indicated by the annotations. Performance is averaged over different compositions of the same pair. We observe that most models fail on "color" based tasks.

abstract rules and hinder generalization. In line with the previous results, SCL-ResNet-18 seems to solve more elementary rules and compositions than the other 3 models.

## 5 Related Work

**Visual reasoning benchmarks** Visual reasoning has been a subject of AI research for decades, and several benchmarks address many relevant tasks. This includes language-guided reasoning benchmarks such as CLEVR [18], which has been extended in its visual composition by recent work [23], physics-based reasoning and reasoning over time dynamics [42, 2]. Abstract visual reasoning benchmarks are more relevant to our work. Raven's Progressive Matrices (RPMs) which were introduced in 1938 [6] are one example used to test human fluid intelligence. Procedural generation techniques for RPMs [39] enabled the creation of the PGM dataset and RAVEN [3, 43]. They also inspired Bongard-Logo [29], a concept learning and reasoning benchmark based on Bongard's 100 visual reasoning problems [4]. Another reasoning dataset, SVRT [12], focuses on evaluating similarity-based judgment and spatial reasoning. Besides these synthetic datasets, real-world datasets were developed with similar task structures to Bongard-Logo and RPM [35, 17]. In this work, we take inspiration from SVRT and develop a more extensive set of rules with careful considerations for the choice of rules and using a novel rule generation method. Finally, Abstract Reasoning Corpus [9] is a general intelligence test introduced with a new methodology for evaluating intelligence and generalization. The numerous problems presented in this benchmark are constructed with a variety of human priors. The unique nature of the task, requiring solvers to generate the answer, and the limited amount of training data render the benchmark difficult for neural network-based methods. We follow a similar approach in our dataset by creating several unique problem templates. However, we restrict the number of samples to a reasonable range to evaluate the sample efficiency of candidate models.

**Compositionality** Compositionality is a highly studied topic in AI research. Although there is agreement over the high-level definition of compositionality; the ability to represent new abstractions based on their constituents and their contexts, there is little consensus on methods for characterizing compositional generalization in neural networks. Several tests for compositionality have been proposed in language [26], mathematics [34], logical reasoning and navigation [5, 21, 32, 41] and visual reasoning [18, 36, 1]. Recent work [16] attempts to identify components of compositionality and proposes a test suit that unifies them. These tests evaluate the model's capacity to manipulate

concepts during inference. Systematicity tests the novel combination of features, akin to CLEVR's CoGenT [18] and C-VQA [1] where novel combinations of shapes and colors introduced in the test set, and localism tests the model's ability to account for context similarly to samples from Winoground [36]. Our work explores compositional generalization from a new perspective; CVR evaluates the model's compositionality while learning novel concepts. A compositional model reuses previously learned concepts to accelerate learning and decomposes complex tasks into elementary components. These aspects of compositionality are tested under settings that employ curricula. Furthermore, we evaluate compositionality over the reasoning operations necessary to solve a given problem. Finally, generating a synthetic dataset allows for evaluating reasoning at high levels of abstraction; groups of objects and scene configurations, as exemplified by tasks in Figure 4.

**Neuroscience/Psychology**   Several theories attempt to propose an understanding of the mechanisms behind visual reasoning. Gestalt psychology provides principles hypothesized to be be used by the visual system as an initial set of abstractions. Another theory describes visual reasoning as a sequence of elemental operations called visual routines [37] orchestrated by higher-level cognitive processes. These elemental operations are hypothesized to form the basis for spatial reasoning, same-different judgment, perceptual grouping, contour tracing and many other visual skills [7]. Evaluating these skills in standard vision models is a recurring subject in machine learning and neuroscience research [19, 24, 30]. To provide a comprehensive evaluation of visual reasoning, it is important to include task sets that require various visual skills within humans' capabilities.

## 6   Discussion and Future Work

In this work, we have proposed a novel benchmark that focuses on two important aspects of human intelligence – compositionality and sample efficiency. Inspired by visual cognition theories [37], the proposed challenge addresses the limitations of existing benchmarks in the following ways: (1) it extends previous benchmarks by providing a variety of visual reasoning tasks that vary in relations and scene structures, (2) all tasks in the benchmark were designed with compositionality prior, which allows for an in-depth analysis of each model's strengths and weaknesses, and (3) it provides a quantitative measure of sample efficiency.

Using this benchmark, we performed an analysis of the sample efficiency of existing machine learning models and their ability to harness compositionality. Our results suggest that even the best pre-trained neural architectures require orders of magnitude more training samples than humans to reach the same level of accuracy, which is consistent with prior work on sample efficiency [22]. Our evaluation further revealed that current neural architectures fail to learn several tasks even when provided an abundance of samples and extensive prior visual experience. These results highlight the importance of developing more data-efficient and vision-oriented neural architectures for achieving human-level artificial intelligence. In addition, we evaluated models' generalization ability across rules – from elementary rules to compositions and vice versa. We find that convolutional architectures benefit from learning all visual reasoning tasks jointly and transferring skills learned during training on elementary rules. However, they also failed to generalize systematically from compositions to their individual rules. These results indicate that convolutional architectures are capable of transferring skills across tasks but do not learn by decomposing a visual task into its elementary components.

While our work addresses important questions on sample efficiency and compositionality, we note a few possible limitations of our proposed benchmark. CVR is quite extensive in terms of the visual relations it contains, but it can always be further improved in its use of elementary visual relations. For example, the shapes could be parametrically generated based on specific geometric features. Hopefully, CVR can be expanded in future work to test more routines by including additional relations borrowed from other, more narrow challenges, including occlusion [19], line tracing [25], and physics-based relations. The rules in the current benchmark are limited to 2 or 3 levels of abstraction to evaluate relations systematically. Similarly, our evaluation methods for sample efficiency and compositionality could be further improved and adapted to different settings. For example, the sample efficiency score is an empirical metric used only for evaluating our benchmark. It requires training all models on all data regimes for the score to be consistent. Although our work is not unique in addressing sample efficiency, its aim is to promote more sample efficient and general models. We hope that the release of our benchmark will encourage researchers in the field to test their own model's sample efficiency and compositionality.

# 7 Acknowledgments

This work was supported by ONR (N00014-19-1-2029), NSF (IIS-1912280 and EAR-1925481), DARPA (D19AC00015), NIH/NINDS (R21 NS 112743), and the ANR-3IA Artificial and Natural Intelligence Toulouse Institute (ANR-19-PI3A-0004). Additional support provided by the Carney Institute for Brain Science and the Center for Computation and Visualization (CCV) and CALMIP supercomputing center (Grant 2016-p20019, 2016-p22041) at Federal University of Toulouse Midi-Pyrénées. We acknowledge the Cloud TPU hardware resources that Google made available via the TensorFlow Research Cloud (TFRC) program as well as computing hardware supported by NIH Office of the Director grant S10OD025181.

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
