# OpenReview forum: "A Benchmark for Compositional Visual Reasoning"
_NeurIPS.cc/2022/Track/Datasets_and_Benchmarks — NeurIPS 2022 Datasets and Benchmarks _

### Official Review · Reviewer_fp5R · 2022-07-18
**Potentially interesting benchmark lacking rigorous design & experiments**

**Rating:** 5
**Confidence:** 4
**Correctness:** Yes.

**Strengths:**

The authors propose an Odd-One-Out reasoning benchmark with more tasks (attributes) and samples that better test model performance on four axes -- sample efficiency, generalization, transfer across task rules, and ability to leverage compositionality. The four identified axes are important to the research community, and findings on the respective models tested are interesting. The dataset is accessible and released with no ethical and social implications.

**Weaknesses:**

Several components of the paper are not defined rigorously or shown as claimed. For example, it is unclear what the “compositionality prior” relied upon in the paper is. All objects have attributes, and most procedurally generated datasets rely on object attributes for generation, such as CLEVR [1]. How is this unique to CVR?

It’s also unclear whether these additional attributes that lead to more tasks truly better test compositional generalization compared to prior visual reasoning tasks, which already test generalization in geometric shapes, color, etc. The setup for testing compositionality (starting L232) is confusing and seems to be overly complicated. A common method of testing compositionality is CLEVR CoGenT [1].

For evaluating models on benchmarks, is there a reason why the models are not pretrained on ImageNet? This would better disentangle the perception and reasoning challenge, and less hamper ViT’s performance. Could this evaluation be added?

It is not clear why the authors' proposed Sample Efficiency Score is better than that of common characterizations of model sample efficiency. Most ML works simply compare performance/accuracy vs. samples and presents numbers at cutoffs.

While the authors attempt to fairly  evaluate Resnet-50 and ViT-small with a similar number of parameters, there is evidence that ViTs suffer from poor initialization [2]. Furthermore, transformer architectures can recognize shapes effectively [3], countering some of the claims in the paper (L280, “We believe that ViT is not capable of learning shape based tasks because of its input processing which splits an image into patches…”). Several of the assertions made about model performance are too strong.

[1] Johnson, J., Hariharan, B., Van Der Maaten, L., Fei-Fei, L., Lawrence Zitnick, C., & Girshick, R. (2017). Clevr: A diagnostic dataset for compositional language and elementary visual reasoning. In Proceedings of the IEEE conference on computer vision and pattern recognition (pp. 2901-2910).
[2] Chen, X., Hsieh, C. J., & Gong, B. (2021). When vision transformers outperform ResNets without pre-training or strong data augmentations. arXiv preprint arXiv:2106.01548.
[3] Naseer, M. M., Ranasinghe, K., Khan, S. H., Hayat, M., Shahbaz Khan, F., & Yang, M. H. (2021). Intriguing properties of vision transformers. Advances in Neural Information Processing Systems, 34, 23296-23308.


**Additional Feedback:**

The paper claims that the dataset is well-positioned to test sample efficiency, generalization, transfer across task rules, and ability to leverage compositionality. Though sample efficiency is convincing, the other three components --- generalization, transfer across task rules, and ability to leverage compositionality --- are not well defined and designed. More rigorous investigation of compositional generalization (compared to prior work such as C-VQA [1] , Winoground [2]) and transfer (ex. with pretrained models, meta-learning) would strengthen the paper.

[1] Agrawal, A., Kembhavi, A., Batra, D., & Parikh, D. (2017). C-vqa: A compositional split of the visual question answering (vqa) v1. 0 dataset. arXiv preprint arXiv:1704.08243.
[2] Thrush, T., Jiang, R., Bartolo, M., Singh, A., Williams, A., Kiela, D., & Ross, C. (2022). Winoground: Probing Vision and Language Models for Visio-Linguistic Compositionality. In Proceedings of the IEEE/CVF Conference on Computer Vision and Pattern Recognition (pp. 5238-5248).


**Clarity:**

Yes. However, parts of the paper are unclear, introduced insufficiently (ex. L74 -- compositionality prior is never defined or explained, L142 -- referring to specific tasks in a different dataset and paper), and confusing (ex. caption for Figure 2 in incorrect order, should move some explanations of the benchmarking process from the appendix into the main test).


**Documentation:**

Yes.

**Ethics:**

No.

**Relation To Prior Work:**

The paper discusses prior visual reasoning benchmarks, but do not discuss prior works or datasets that measure model performance on the four axes that the benchmark tests (sample efficiency, generalization, transfer across task rules, and ability to leverage compositionality). Importantly, it seems that one of the main differences between CVR and visual reasoning benchmarks is the unique combination of the samples it contains and the tasks it captures. However, it is not convincing that this is a benchmark that needs to exist (in axes such as testing compositional generalization), instead of creating more tasks in the style of RAVEN & SVRT or more training samples in ARC. Paper also seems to miss some literature in odd-one-out tasks, for example G1-set & S1-set [1].

[1] Mańdziuk, J., & Żychowski, A. (2019, July). Deepiq: A human-inspired ai system for solving iq test problems. In 2019 International Joint Conference on Neural Networks (IJCNN)(pp. 1-8). IEEE.


**Summary And Contributions:**

Proposed CVR dataset for Odd-One-Out visual reasoning that tests sample efficiency, generalization, transfer across task rules, and ability to leverage compositionality. CVR contains 103 unique task rules over distinct scene structures generated procedurally. The paper analyzes and evaluates convolutional and transformer-based models on the proposed task and stated axes.

*EDIT: I thank the authors for clarifying some of the dataset details. I have raised my score slightly, though I still believe that the paper lacks clarity and is not ready for publication.*

---

> ### Author Response · Authors · 2022-08-17
> **Response to reviewer fp5R [Part 1]**
>
> Thank you for taking the time to review our paper and provide helpful feedback. Your perceptive comments have helped us to extend and refine the paper. We incorporated several changes to the manuscript and addressed your concerns as follows.
>
> [**Lack of rigorous definitions, Comparison to CLEVR, confusing compositionality tests**] We acknowledge the lack of a proper definition for compositionality or an explanation of its use in the benchmark. We thank the reviewer for the insightful references. We enriched the dataset description section with more information on compositionality and the related work section with comparisons to relevant papers. Our paper evaluates whether models leverage compositionality in their learning i.e. do the models learn novel concepts in a compositional manner? To answer this question, we investigate whether models learn a concept (a visual reasoning task) faster if they have already learned its components and whether a model trained on a new task decomposes it into its constituent parts. Although our quantitative evaluations of compositionality are only empirical, we believe they provide insights into important aspects of compositionality, particularly in the domain of visual reasoning
>
> [**Imagenet pre-trained models**] We provide new results on Imagenet pre-training. We observe that ImageNet pre-trained models perform better than randomly initialized ones but still worse than SSL pre-trained models. To provide a less biased comparison with humans, we finetuned the visual encoder of CLIP; a model trained to annotate images with language, on our dataset. We believe that image annotation is a task that requires visual reasoning capabilities. It is also a natural task, akin to scene understanding that humans perform regularly. We find that ViT-Base, initialized with CLIP visual encoder weights, achieves high performance, close to SSL pre-trained ResNet-50 in the 1,000 samples data regime. We believe this performance mainly results from the image annotation pre-training and a large number of parameters compared to ViT-small since ResNet-50 initialized with CLIP visual encoder weights reaches a lower performance than SSL pre-trained ResNet-50.
>
> | Model  | 20 | 50 | 100 | 200 | 500 | 1000 | SES | AUC |
> | --- | --- | --- | --- | --- | --- | --- | --- | --- |
> | resnet 50 (Imagenet) | 32.0 / 2 | 35.1 / 5 | 39.0 / 9 | 43.8 / 13 | 57.7 / 48 | 69.5 / 48 | 43.4 | 46.2 |
> | vit small (Imagenet) | 27.9 / 2 | 28.2 / 1 | 28.6 / 2 | 30.0 / 0 | 35.6 / 5 | 47.2 / 24 | 31.7 | 32.9 |
> | vit base (clip) | 31.1 / 1 | 37.4 / 7 | 43.9 / 14 | 56.0 / 30 | 68.9 / 48 | 78.8 / 62 | 48.9 | 52.7 |
> | resnet 50 (clip) | 28.7 / 0 | 32.0 / 2 | 40.8 / 11 | 46.9 / 18 | 59.7 / 40 | 74.4 / 53 | 43.7 | 47.1 |
>
> [**Why propose a sample efficiency score ?**] We present accuracy at different data regimes and believe that it is difficult to compare models when they have different scores at different data regimes. For example, when comparing SSL pre-trained ViT-small to CLIP pre-trained ViT-base, we find that the former has a higher performance in lower regimes whereas the latter achieves better performance in the higher data regimes. SES provides a comparison that considers all data regimes and favors models with high performance at low data regimes.. Beyond a comprehensive comparison, the score simplifies the evaluation of models on this benchmark and encourages research on sample efficient models.

---

> > ### Author Response · Authors · 2022-08-17
> > **Response to reviewer fp5R [Part 2]**
> >
> > [**Unfair evaluation of ViTs**] We agree and will revise/soften the claims about the generalizability of the results obtained with ViTs on our benchmark. However, as we mentioned in the supplementary material, we made every effort to give each model the best chance to solve the tasks. In preliminary experiments, we tried different methods for selecting the odd-one-out and performed hyperparameter search for all models using joint rule learning and 500 samples per rule. We also verify that results do not vary for different random seeds. The training schemes and hyperparameters used in the main experiments are the best we could achieve within our computing budget. Furthermore, following the reviewer’s advice, we applied novel optimization techniques for ViTs with a range of learning rates and weight decays and observed that they do not improve ViT’s performance in any data regime in the joint rule learning setting. In our opinion, these results reflect the true capabilities of the model.
> >
> > | Model  | 20 | 50 | 100 | 200 | 500 | 1000 | SES | AUC |
> > | --- | --- | --- | --- | --- | --- | --- | --- | --- |
> > | ViT small | 26.4 - 1 | 26.7 - 1 | 27.5 - 1 | 27.8 - 1 | 28.8 - 2 | 32.2 - 3 | 27.9 | 28.2 |
> >
> > [**Lack of clarity**] We acknowledge the lack of clarity. We edited the manuscript accordingly. We expand upon the compositionality prior in the dataset description section.
> >
> > [**Prior work discussion - motivation for creating the dataset - evaluating generalization, transfer and compositionality**] We have complemented the related works section with a paragraph that compares CVR to relevant work on evaluating compositionality and we cite the referenced paper in the Odd-One-Out task description. Furthermore, we highlight the novelty of CVR in the field. As we mentioned in an earlier comment, our paper evaluates whether models leverage compositionality in their learning process and the compositionality prior employed for creating the dataset is crucial for this evaluation. We find that expanding other benchmarks, such as ARC, SVRT, and RAVEN is not sufficient to evaluate this aspect of compositionality because they are not built using the same compositionality prior. Finally, although our quantitative evaluations of compositionality are only empirical, we believe they provide insights into important aspects of compositionality that are not well explored in the literature.
> >
> > **Final remake** In conclusion, we found this review helpful and believe this rebuttal process has resulted in substantial improvements to our submission. Thank you again for this contribution, and please feel invited to engage with us if you have further questions. If you agree that our revisions address your concerns about clarity, methodology and novelty, we would appreciate it if you could consider raising the score.

---

> > > ### Comment · Reviewer_fp5R · 2022-08-24
> > > **Response**
> > >
> > > Please clarify the difference between CVR and SVRT more concretely. As mentioned in my above review, this part (L150, previously L142) is very unclear and references specific tasks. I would recommend a table comparing the two datasets (and other related datasets).

---

> > > > ### Author Response · Authors · 2022-08-27
> > > > **Response to reviewer fp5R**
> > > >
> > > > We thank the reviewer for engaging further in the discussion. If we understood correctly, the reviewer is concerned about the clarity of the **technical** differences between CVR and SVRT. We will provide a clearer description of SVRT and its differences with CVR in addition to a figure with examples of SVRT tasks.
> > > >
> > > > We would like to remind the reviewer that we made an effort to highlight the novelty and the significance of CVR in comparison to other datasets. The most notable differences are the number of unique rules (103 for CVR and 23 for SVRT) and the task setup (Odd-One-Out for CVR and binary classification for SVRT) which are explained in Figure 1 and the dataset description L150-L163. These sections compare CVR to other datasets as well.
> > > >
> > > > However, we do not provide detailed descriptions of all comparable benchmarks because it is not necessary for understanding the main focus of the paper.
> > > >
> > > > The following table lists differences between relevant benchmarks and CVR.
> > > >
> > > > | Dataset  | N tasks | N samples | Task Setup | Generalization Test | Transfer Learning Test | Compositionality Test |
> > > > | --- | --- | --- | --- | --- | --- | --- |
> > > > | SVRT | 23 | $\infty$ | binary classification | | | | |
> > > > | PGM | 25 | 48000 | RPM | x | | |
> > > > | RAVEN | 28 | 40000 | RPM | x | | |
> > > > | ARC | 1000 | 3 | image generation | | | |
> > > > | CVR | 100 | $\infty$ | odd-one-out | x | x | x |

---

### Official Review · Reviewer_ehHC · 2022-07-27
**Review on A Benchmark for Compositional Visual Reasoning**

**Rating:** 6
**Confidence:** 3

**Strengths:**

* The authors provide a benchmark that creates tasks with a wide variety of underlying abstract rule conditions constructed from some primitive rule conditions; thereby leading to a compositional prior in the data generating process.
* The authors show that models which are trained jointly on all tasks perform much better than those trained on tasks individually. It would be interesting to see an experiment that validates or invalidates this trend with increasing model sizes, whether this trend exists irrespective of the size of the model or only for models that are larger than a certain threshold.
* The work provides clear evidence of benefits of SSL-based pretraining for visual reasoning tasks.

**Weaknesses:**

* Maybe I misunderstood it, but the benchmark does not have any notion of out-of-distribution generalization based testing? Given the structure of the dataset, it would be nice to have this functionality present as well.
* Given that the models are trained in extremely low data regime in Table 1, it would be nice to have some estimate of standard error / deviation to understand if the difference between the algorithms is statistically significant since low data regime might lead to high deviation in results.
* There are a lot of details missing on how the dataset / tasks are generated. For example, an image of a few different tasks with compositional abstract underlying rule should be there in the main paper to make it easier to understand the kind of tasks that are being considered. There should also be additional details on how a scene is generated from the scene-graph, eg., how are two objects generated such that their contours are touching or how is it ensured that objects do not overlap, etc. Such details are missing from the draft.

**Additional Feedback:**

More details on how the dataset is actually created would be quite helpful!

**Clarity:**

* I could not understand Figure 3 completely. What is the reference and odd rule? How is the fourth frame violating the reference rule? Is there any need for the odd rule when any rule not equal to reference rule can be considered as an odd rule?
* Is it possible to have model performance without any curriculum on Figure 6b too to serve as a baseline to compare against?
* In Figure 7, what do the columns represent? What are the "group of composition rules?"? Can you provide an example on this to understand?

**Correctness:**

* The claim that a lot of AI systems are less data efficient when compared to humans might be true but is not substantiated by the self-supervision based experiments. This is because while self-supervision does learn some knowledge about the visual semantics of the task, humans are introduced to much more complex visual stimuli as well as by actually acting in the world, they also a-priori develop knowledge about various kinds of reasoning. It could be that this a-priori abundance of information makes them more data efficient for specific tasks, and not the actual architecture itself. A closer step to this analysis might be using a large pre-trained network and finetuning it on the tasks presented; but even that might not be enough to substantiate this claim unless quantitative numbers demonstrate that the pre-trained system and humans both were "pre-trained" on the same amount and kind of information.
* The comparison between Figure 6a and 6b doesn't seem fair since in 6a the models are finetuned on difficult compositions while in 6b they are not finetuned on elementary relations and are just expected to perform well zero-shot. Does the same trend hold when you finetune in 6b or do 6a zero-shot?

**Documentation:**

The authors provide a URL containing the codebase for the benchmark as well as documentation on its usage.

**Ethics:**

I do not see any ethical concerns with the work.

**Relation To Prior Work:**

* The idea of better sample complexity and inductive biases for better out-of-distribution generalization has been presented in a number of works and it would be nice to have a discussion along these lines to better understand how one can come up with systems that can do well in the provided tasks. Some examples of works that study these are:
  * Kerg et. al 2022 On Neural Architecture Inductive Biases for Relational Tasks
  * Mittal et. al 2021, 2022 Compositional attention: Disentangling search and retrieval and Is a Modular Architecture Enough?
  * Rahaman et. al 2021 Dynamic inference with neural interpreters
  * Goyal et. al 2019 Recurrent independent mechanisms
  * Bergen et. al 2021 Systematic Generalization with Edge Transformers

**Summary And Contributions:**

The authors provide a benchmark aimed at testing the ability for compositional generalization in visual reasoning based settings. To this end, they provide a wide variety of tasks with different underlying relations and perform individual as well as joint training over the different tasks. The work also covers a self-supervised pretraining baseline, which is more comparable to human performance as humans generally have a well developed visual cortex before they are tasked with such reasoning problems. Finally, analysis is done over task difficulties, sample efficiencies of models, ability for compositional generalization as well as trends in the presence and absence of curriculum over the tasks.

---

> ### Author Response · Authors · 2022-08-17
> **Response to reviewer ehHC [Part 1]**
>
> We appreciate the reviewer for the careful review and great pointers to related works. We have edited the manuscript accordingly and addressed the reviewer's concerns as follows:
>
> [**No OOD Generalization testing**] The OOD generalization setting and results were provided in the supplementary material (Table 2). They show that most models generalize poorly to out of distribution data.
>
> [**Standard deviation in low data regimes**] We agree with the reviewer on the importance of error bars in figures. We were not capable of adding them because of the high computational cost of running the experiments with more seeds. In preliminary experiments, we performed a hyperparameter search and tried different random seeds. We noticed that the performance is robust to the random seed. Nevertheless, we provide new results that now include error bars on the lowest and highest data regimes (20 and 1000 samples) in the joint rule learning setting. For each model, we provide results over four training runs with random seeds. The results show low standard deviations reaching a maximum of 0.31% for the 20 samples data regime and 2.06% for the 1000 samples data regime. We will try our best to provide results for the other data regimes in the final version of the paper. We hope these results convince the reviewer of the robustness of the results in the paper.
>
> | Model                   |  20           | 1000     |
> | -------                 | -----              | -----         |
> | ResNet50         | 27.2 (+/- 0.29) | 62.7 (+/- 2.06) |
> | ViT-small         | 27.3 (+/- 0.17) | 32.5 (+/- 0.77) |
> | SCL             | 25.5 (+/- 0.27) | 46.5 (+/- 1.05) |
> | WReN         | 26.8 (+/- 0.20) | 42.8 (+/- 0.62) |
> | SCL-ResNet18     | 26.8 (+/- 0.31) | 64.9 (+/- 0.72) |
>
> [**Missing details of the generation process**] We acknowledge the lack of details on certain aspects of the generation process. We provide a clearer explanation in section 3 (dataset description) and the related works section of the manuscript. More details are also available in section 2 of the supplementary materials.
>
> [**Correctness of the comparison to human data**] We acknowledge that humans perform visual reasoning tasks regularly, which gives them an advantage over these models. It is difficult to reproduce the diverse human visual experience in machine learning models. We choose to pre-train models with self-supervision because this method was shown to be helpful in finetuning on several visual tasks. It might even endow models with visual capabilities due to the nature of the task (selecting similar images up to random transformations). To provide a less biased comparison, we finetune the visual encoder of CLIP, a model trained to annotate images with language, on our dataset. We believe that image annotation is a task that requires visual reasoning capabilities. It is also a natural task that humans perform regularly, akin to scene description. We find that ViT-Base, initialized with CLIP visual encoder weights, achieves high performance, close to SSL pre-trained ResNet-50 in the 1000 samples data regime. We believe this performance mainly results from the image annotation pre-training and the large number of parameters compared to ViT-small since ResNet-50 initialized with CLIP visual encoder weights reaches a lower performance than SSL pre-trained ResNet-50. Nevertheless, all models do not reach the human-level sample efficiency. We will include these new results in the manuscript. We understand that pre-training models on most visual tasks do not provide them with a visual experience equivalent to humans, but we hope that the new results provide the reviewer with more insight into this comparison and supporting evidence for our hypothesis.
>
> | Model  | 20 | 50 | 100 | 200 | 500 | 1000 | SES | AUC |
> | --- | --- | --- | --- | --- | --- | --- | --- | --- |
> | resnet 50 (Imagenet) | 32.0 / 2 | 35.1 / 5 | 39.0 / 9 | 43.8 / 13 | 57.7 / 48 | 69.5 / 48 | 43.4 | 46.2 |
> | vit small (Imagenet) | 27.9 / 2 | 28.2 / 1 | 28.6 / 2 | 30.0 / 0 | 35.6 / 5 | 47.2 / 24 | 31.7 | 32.9 |
> | vit base (clip) | 31.1 / 1 | 37.4 / 7 | 43.9 / 14 | 56.0 / 30 | 68.9 / 48 | 78.8 / 62 | 48.9 | 52.7 |
> | resnet 50 (clip) | 28.7 / 0 | 32.0 / 2 | 40.8 / 11 | 46.9 / 18 | 59.7 / 40 | 74.4 / 53 | 43.7 | 47.1 |

---

> > ### Author Response · Authors · 2022-08-17
> > **Response to reviewer ehHC [Part 2]**
> >
> > [**Compositionality experiments**] Figures 6a and 6b try to answer different questions; in 6a, we seek to understand whether providing the model with a curriculum (elementary tasks) will help it learn with fewer samples in the curriculum condition. In 6b, however, we ask whether a model learns the elementary tasks while learning the composition, i.e., is it decomposing the task? Answering these questions requires different evaluation methods. To give more details, we consider the composition rules objectively harder than elementary rules. A model pre-trained on two elementary rules is unlikely to solve the composition zero-shot because it has never been trained on a composition task and has no notion of compositionality. It could learn the composition task faster if it relies on prior abstractions built during the elementary task training.
> > Similarly, if a model learns a composition, it should not require finetuning to perform the elementary tasks zero-shot. To clarify our point, we provide 0-shot evaluation results in the supplementary material. These results show that all models pre-trained with elementary rules fail at their compositions when evaluated in a 0-shot manner. Due to a lack of computing resources and time, we were not able to provide finetuning results in 6b yet. However,  we seek to include them in the final version of the paper.
> >
> > [**Unclear Figure 3**] We revised the manuscript to clarify the explanation of the problem generation process. We modified figure 3 and provide more details in the dataset description. Tasks are designed such that they require specific reasoning operations to solve. The generation process relies on the definition of a reference rule that three images will respect and an odd rule that violates the reference rule in a manner that targets elementary rules. For example, considering a composition rule based on color and numerosity, the odd-one-out rule is similar to the reference rule. Still, it differs such that a problem solver needs to reason exclusively over colors and numbers. Choosing arbitrary odd rules renders the task arbitrary. In figure 3, the reference rule is “among 4 objects, those with the same shape have the same size”. It is a composition of the “shape” and “size” elementary relations. The odd rule keeps the same number of objects, 2 pairs of similar shapes and 2 pairs of similar sizes. However, in the odd rule, the objects with similar shapes have different sizes. The problem solver has to reason over the similarity of shapes and similarity of sizes to solve problem samples. If the odd rule had 5 objects instead of 4, the problem solver could simply count the objects to find the outlier.
> >
> > [**Change figure 6b**] We will change the graph such that it takes into account the performance and not only the difference in performance. We find that figure 6a allows a more direct qualitative comparison of models using the difference in accuracy between the two conditions (curriculum vs. control). However, the new plots included in the main paper provide new insights; due to the varying difficulty of the tasks, accuracy in 6a either 1- spans all 25%-100% across data regimes, 2- is consistently low for difficult tasks or 3- consistently high levels for easy tasks. The analysis is most interesting for the first case since it shows best which models benefit from elementary tasks pre-training at intermediate data regimes. For example, the elementary task pre-training allows ResNet50 to perform several tasks with fewer training samples. This is highlighted by the large difference in accuracy at the 200 and 500 data regimes in several tasks. These new results inspired us to propose a quantitative compositionality score for this evaluation setup. The score computes the maximum difference in accuracy across data regimes (at the data regime for which we can observe the most benefit from elementary task pre-training) and averages this value across tasks. We observe that the qualitative advantage for SCL-ResNet-18 is consistent with the quantitative evaluation. SCL and ResNet-50 also benefit from elementary task pre-training more than WReN and ViT-small. These new results are added to the supplementary material in table 2.
> >
> > | Model                   | average accuracy difference     |
> > | -------                 | -----        |
> > | ResNet 50         | 12.1 |
> > | ViT small         | 2.61 |
> > | SCL            | 12.6 |
> > | WReN         | 6.75 |
> > | SCL-ResNet 18     | 23.1 |

---

> > > ### Author Response · Authors · 2022-08-17
> > > **Response to reviewer ehHC [Part 3]**
> > >
> > > [**Explain figure 7**] We acknowledge that figure 7 is hard to interpret. It was difficult to create a visualization that conveys information about the elementary components of all tasks. In figure 7, the top row, separated by a dark line, shows the performance of elementary rules; the corresponding tasks are aligned on the diagonal. The triangle shows performance on compositions of pairs of rules. The pairs of rules are indicated by the annotations on the diagonal. Cells on the diagonal are complex tasks that require reasoning over only one relation. For example, the average accuracy on “contact” and “shape” compositions is in the cell on the top right of the triangle. The term "group of composition rules" refers to rules that are based on the same pair of elementary rules. As shown in the supplementary material Figure 1, for example, there are 3 different color-shape rules. Accuracy in each cell is averaged across these tasks. We will improve the explanation provided for this figure in the manuscript.
> > >
> > > [**discussion about sample complexity and OOD generalization**] We thank the reviewer for highlighting this point of the discussion and providing insightful references. Although the paper does not focus on model design, it is important to discuss the topic for future work. We provide a discussion of promising inductive biases for visual reasoning in the supplementary material’s section 8.
> > >
> > > [**more details on the dataset creation**] This concern is shared by several reviewers. We will make changes to the description of the generation process and provide more details in the supplementary material.
> > >
> > > **Final remark** In summary, we would like to thank the reviewer again for the insightful comments. We believe this rebuttal process has largely improved our submission. We would be eager to interact with you if you have additional questions. We hope that our responses will address your concerns. We would appreciate it if the reviewer could consider raising the score.

---

> > > > ### Comment · Reviewer_ehHC · 2022-08-29
> > > > **Response**
> > > >
> > > > I appreciate the authors' response but will choose to keep my score as is. I would also suggest the authors to use bigger models in their experiments; I saw they use MLPs with 2048 hidden dimensions but the output dimension as just 128. I think it would be nice to have analysis up to output dimension 512 just to see how well can the task be solved by current architectures when they are not so bottle-necked.

---

> > > > > ### Author Response · Authors · 2022-08-29
> > > > > **Response to reviewer ehHC**
> > > > >
> > > > > We thank the reviewer for participating in the discussion and providing more feedback. We hoped that our responses to questions and modifications of the manuscript and SI had addressed all the reviewer's concerns. We would have been eager to address the remaining concerns if the discussion period had not ended.
> > > > >
> > > > > Although the MLP output size analysis would be interesting, we believe the models are not bottlenecked since they can solve most tasks with 128 dimensions (ResNet50 achieves ~93% in the 10k data regime). Furthermore, the MLP output size was among the hyperparameters we included in the search and did not impact performance beyond the 128 dimensions.

---

> > > > > > ### Comment · Reviewer_ehHC · 2022-08-29
> > > > > > **Response**
> > > > > >
> > > > > > Thanks! Since increasing the MLP output size does not impact performance much, it clarifies all of my doubts. I have updated the score accordingly.

---

> > > > > > > ### Author Response · Authors · 2022-08-29
> > > > > > > **Response to reviewer ehHC**
> > > > > > >
> > > > > > > Thank you for your interacting with us. We appreciate that you have adjusted your score given the additional results and clarifications. Your feedback has significantly improved the presentation of our paper.

---

### Official Review · Reviewer_buL9 · 2022-07-28
**The paper proposes a new benchmark for visual reasoning tasks leveraging compositionality and motivates research into the investigation of the modern neural nets.**

**Rating:** 6
**Confidence:** 5
**Correctness:** Yes.
**Clarity:** The paper is well written.

**Strengths:**

The strengths of the paper are :-
1. The proposed method creates numerous tasks of varying levels of difficulty for assessment of visual reasoning using elementary rules and their composition.
2. The dataset can be synthesized through a program set by rules and multiple levels of abstraction can be constructed.
3. They also conduct analysis of the sample efficiency and evaluate on tasks of composition which is significant.
4. The benchmark makes an important case to highlight the inability of modern neural nets in visual reasoning tasks and has potential to improve their interpretability.


**Weaknesses:**

The weakness are as follows :-
1. The proposed task is of one kind - odd-one-out rule. What about task such as visual completion of shapes based on pattern recognition? More varied tasks could be devised.
2. The reasoning evaluation is restricted to accuracy of the task only and there is no analysis on the representations learnt by the neural nets or visual explainablity of the task.




**Additional Feedback:**

The claim on learning of composition for transformers vs convolution architectures need further assessment, as both differ in optimization schemes under same data regime.

**Documentation:**

Yes.

**Ethics:**

I do not see any ethical concerns.

**Relation To Prior Work:**

The comparison to prior work is shown.

**Summary And Contributions:**

The paper proposes a novel benchmark for assessment of visual reasoning of modern neural net architectures. It proposes tasks by creation of basic abstract rules and use compositionality on them to create additional rules. In addition, they propose measures of sample efficiency and test ability of model generalization across different rules. Modern neural nets are evaluated and also compared against human baseline for data efficiency and overall accuracy of tasks.

---

> ### Author Response · Authors · 2022-08-17
> **Response to reviewer buL9**
>
> We thank the reviewer for the interest in the benchmark, as well as for the insightful comments. We have revised our manuscript accordingly and addressed the reviewer's concerns as follows:
>
> [**Other task frameworks than Odd-One-Out**] In this work, we focus on the odd-one-out task because (1) it is flexible (adaptable to any arbitrary rules) and (2) it does not require the design of special architectures (as we show in our experiments, a simple architecture design is enough to solve several tasks). This choice simplifies the task design and allows us to focus on evaluating the reasoning and visual capacities of SOTA models. However, we believe that the generation process can be adapted for a variety of tasks, such as binary classification, Raven's Matrices, pattern completion as the reviewer mentions, and several novel tasks can be created in each task framework.
>
> [**Lack of explainability analysis**] We agree with the referee that it is important to gain a deeper, mechanistic understanding of the computations required to solve this task. We followed the reviewer’s suggestion and tried a few representative explainability methods (attribution methods: smoothgrad, saliency, and integrated gradients) to explain model decisions. Results are shown in figure 10 of the supplementary material. Unfortunately, we have found that they do not provide real insights into the models. Since models have to use information extracted from all objects in the scene in all tasks and the background remains a uniform white color, the gradient maps focus on object contours and their surroundings. We added a few representative maps in the supplementary material. Visual reasoning in general is a high-level cognitive task and it will require more sophisticated explainability methods to analyze.
> In addition, we also provide new analyses of pairwise differences of models in different settings in the supplementary material, figure 9. We try to understand which tasks pre-training facilitates and which ones are facilitated by the inductive bias of the model.
>
> [**Unfair comparison because of different optimization schemes**] We agree with the reviewer on the importance of utilizing the best optimization schemes to provide a fair comparison between models. In preliminary experiments, we tried different methods for selecting the odd-one-out and performed a hyperparameter search for all models using joint rule learning and 500 samples per rule. We also verify that results do not vary for different random seeds. The training schemes and hyperparameters used in the main experiments are the best we could achieve with our computing budget. Furthermore, following reviewer fp5R’s advice on optimizations schemes, we applied novel optimization techniques for ViTs [1] with a range of learning rates and weight decays and observed that they do not improve ViT’s performance in any data regime in the joint rule learning setting. To our understanding, these results reflect the true capabilities of the model. We encourage the reviewer to propose changes to our presentation or additional experiments to strengthen the paper.
>
> | Model  | 20 | 50 | 100 | 200 | 500 | 1000 | SES | AUC |
> | --- | --- | --- | --- | --- | --- | --- | --- | --- |
> | ViT small | 26.4 - 1 | 26.7 - 1 | 27.5 - 1 | 27.8 - 1 | 28.8 - 2 | 32.2 - 3 | 27.9 | 28.2 |
>
> **Final remark**: We thank the reviewer for helping us improve our submission. Please feel invited to leave more comments in case you have additional questions. We hope that our responses will address your concerns on the analyses and methodology. We would appreciate it if the reviewer could consider raising the score.
>
> [1] Chen, X., Hsieh, C. J., & Gong, B. (2021). When vision transformers outperform ResNets without pre-training or strong data augmentations. arXiv preprint arXiv:2106.01548.

---

### Official Review · Reviewer_FKHa · 2022-07-28
**A beneficial dataset alongside a generation method for comparing models on Visual Reasoning**

**Rating:** 8
**Confidence:** 3

**Strengths:**

- Besides the data splits provided (including a generalization split), the method presented for generating the dataset problems could be easily extended to incorporate different tasks to compare other characteristics of interest or a diverse set of visual relations.
- The thorough comparison of SOTA models includes sample efficiency for different training sizes and also evaluates the ability to use compositionality via contrasting the performance of training the models using individual or joint tasks and the capability of transferring the knowledge acquired on training to unseen tasks.
- The evaluation of the task difficulty according to the distinct elementary relations of the objects.
- The incorporation of a hybrid model that combined a convolutional backbone with a visual reasoning model against either purely deep learning or purely visual reasoning models.


**Weaknesses:**

- On the use of the self-supervised pre-training on ImageNet, the authors claim that this is performed to dissociate the feature learning from the reasoning learning. Nevertheless, the average human performing these tasks has almost surely already performed any other visual reasoning task different from discriminating objects. It could be interesting to see what happens when using another visual reasoning pretraining like Raven or PGM.
- The human data to compare the results shown in this work is not provided. It would be interesting to include that data. Some minor population data like the age of the subjects could be also included.


**Additional Feedback:**

Some minor comments and typos:
- Is the AUC reported equally averaged across each training scenario or is weighted like the SES metric?
- On L165, “... (the models are trained) jointly on several tasks”, do you mean all tasks or a sample of them?
- Figures 1 and 2 are never referenced in the text
- Figure 7, the last model I believe is supposed to be SCL-ResNet 18 instead of SCL-ResNet 10
- Supplementary L33 “Genralization”
- Supplementary Table 1 SCL-ResNet 18 line is missing the units (M)


**Clarity:**

The paper is well-written, with clear details of the limitations and contributions of the present work. Still, the number of characteristics compared between models made the work a little difficult to follow.

**Correctness:**

The construction of the dataset is correct and the consequent evaluation of the models is also fine. Limitations of the work are included and discussed properly. The claim of 50 times more samples is needed for neural architectures to compete than a human seems a bit of an overstatement.

**Documentation:**

The authors provide a Github repository and a very complete datasheet. On the repository, there is the information needed to reproduce the results with the exception of human behavioral data.

**Ethics:**

I find no ethical concerns in the present work.

**Relation To Prior Work:**

The authors describe extensively how their work compares against previous work. They compare where this dataset stands on the visual reasoning benchmarks in relation to the number of tasks and number of samples.


**Summary And Contributions:**

In this work, the authors present a benchmark for comparing the performance of models on visual reasoning with a focus on assessing sampling efficiency and compositionality. The authors contribute with a dataset, a method for generating visual reasoning problems, and an extensive comparison of different state-of-the-art vision models including a new model

---

> ### Author Response · Authors · 2022-08-17
> **Response to reviewer FKHa**
>
> We greatly appreciate the comments, suggestions and corrections proposed by the reviewer. The perceptive feedback has helped us to extend and refine the paper. We address the concerns in detail as follows.
>
> [**SSL comparison to humans**] We would like to clarify that we used sample images from CVR to pre-train models with self-supervision. However, we also provide new results on Imagenet pre-training. We observe that ImageNet pre-trained models perform better than randomly initialized ones but still worse than SSL pre-trained models. All models still do not reach human-level performance.
> Even though we make an effort to embed standard vision models with enough visual experience to compare them fairly with humans, we acknowledge that the comparison could be fairer since humans have the advantage of performing visual reasoning tasks regularly. It is difficult to reproduce the diverse human visual experience in machine learning models. That said, (self)-supervised learning of object features is considered an important factor in shaping representations for semantic reasoning (e.g., Rogers & McClelland, 2004 [1]). Nevertheless, to provide a less biased comparison, we finetuned the visual encoder of CLIP, a model trained to annotate images with language, on our dataset. We believe that image annotation is a task that requires visual reasoning capabilities. It is also a natural task, akin to scene understanding that humans perform regularly. We find that ViT-Base, initialized with CLIP visual encoder weights, achieves high performance, close to SSL pre-trained ResNet-50 in the 1,000 samples data regime. We believe this performance mainly results from the image annotation pre-training but also the large number of parameters in ViT-base compared to ViT-small since ResNet-50 initialized with CLIP visual encoder weights reaches a lower performance than SSL pre-trained ResNet-50. Nevertheless, all models do not reach the sample efficiency of human participants. We will include these new results in the supplementary material of the paper.
>
> | Model  | 20 | 50 | 100 | 200 | 500 | 1000 | SES | AUC |
> | --- | --- | --- | --- | --- | --- | --- | --- | --- |
> | resnet 50 (Imagenet) | 32.0 / 2 | 35.1 / 5 | 39.0 / 9 | 43.8 / 13 | 57.7 / 48 | 69.5 / 48 | 43.4 | 46.2 |
> | vit small (Imagenet) | 27.9 / 2 | 28.2 / 1 | 28.6 / 2 | 30.0 / 0 | 35.6 / 5 | 47.2 / 24 | 31.7 | 32.9 |
> | vit base (clip) | 31.1 / 1 | 37.4 / 7 | 43.9 / 14 | 56.0 / 30 | 68.9 / 48 | 78.8 / 62 | 48.9 | 52.7 |
> | resnet 50 (clip) | 28.7 / 0 | 32.0 / 2 | 40.8 / 11 | 46.9 / 18 | 59.7 / 40 | 74.4 / 53 | 43.7 | 47.1 |
>
> [1] Rogers, Timothy T., and James L. McClelland. Semantic cognition: A parallel distributed processing approach. MIT Press, 2004.
>
> [**Providing human data**] We include a table detailing the results of the experiment in the GitHub repository and give more details in the description of behavioral experiments in the supplementary material. https://github.com/aimzer/CVR/tree/main/behavioral_experiments
>
> [**Strong claim about sample efficiency**] Although the paper claims that humans are 50 times more sample efficient than ML models, this claim is only reflected by experiments on CVR. It is not a statement that we generalize beyond our experimental setting. Our statement aims to highlight the large gap between humans and machines in sample efficiency. We will clarify this statement in the paper.
>
> [**Clarity**] We hope that the new modifications to the manuscript will bring clarifications. If the reviewer still finds difficulties following the paper, we would be happy to know which sections need further clarification.
>
> [**additional feedback**] We corrected the typos and errors highlighted in the additional feedback. The AUC is equally averaged across data regimes, unlike SES. We clarified these details in the main text. In the joint rule learning setting, models are trained on all tasks.
>
> **Final remark**: In summary, we thank you again for investing the time and effort to review our paper and for the helpful comments that helped us improve the submission. We hope that our responses will address your concerns on the comparison to humans and invite you to engage with us if you have more questions. We would appreciate it if you could consider raising the score.

---

### Official Review · Reviewer_QftC · 2022-07-28
**Review for A Benchmark for Compositional Visual Reasoning**

**Rating:** 5
**Confidence:** 3
**Correctness:** Yes.
**Clarity:** Yes.

**Strengths:**

1. Interesting method for generating the odd-one-out dataset

The paper proposes a well-defined generation process of the odd-one-out dataset. In one example, three images will share a set of rules while one image will not; meanwhile, random parameters are used to insure all images are “dissimilar” on the surface. This also allows targeted test at models’ ability to reason over certain properties (e.g., Figure 7).


2. Interesting observations

The paper presents a study on how models with different architectures (ResNet v.s. ViT) learn compositional visual reasoning. They have made several interesting observations:

a. Under low-data regime, ViT is not as data-efficient as CNN.

b. ViT does not learn shapes well presumably due to the tokenization process. Such an issue is not well noticed on real-world benchmarks.




**Weaknesses:**

1. A rather vague definition of compositionality
The word compositionality is used rather lightly. I would have appreciated more discussion on how this dataset presents a compositional challenge and how it compares to the compositional challenge in prior work (e.g., Compositionality decomposed: how do neural networks generalise?).

For example, in CLEVR, compositionality means generalization to novel combinations of object categories and attributes. In this very paper, it means solving problems requiring multiple rules. Does this present a unique compositionality challenge? In CLEVR, it seems the compositionality challenge mainly lies in learning compositional representation (i.e., correctly identify and represent novel objects). It is the same in this paper?


2. Discussion with respect to related work
One of the selling points of the dataset is its limited size and focus on data efficiency. However, one could also simply downsample the training set of prior datasets such as SVRT. I am not sure the core novelty of this dataset compared to prior datasets (I guess this is also related to my question 1).


3. Limited size
Why stop at generating 1000 examples per rule? Would it be possible to generate more examples and test how many examples are needed for the models to reach high accuracy? This could be informative for future development.


**Additional Feedback:**

1. Why join-ViT underperforms ind-ViT (Table 1)


2. How do we model 4 images with the model? Do we concat the 4 images into a large image and feed it into the model?


3. The title is a bit too broad

**Documentation:**

Yes.

**Ethics:**

N/A.

**Relation To Prior Work:**

See weakness.

**Summary And Contributions:**

The paper proposes to create an  odd-one-out dataset  for visual reasoning, where each example has 4 images and 3 of the images obey one set of rules while one image does not. The paper describes a well-thought-out process to generate such data.

Experiments show that this dataset indeed presents a challenge to current models under a low-data setting (1000 images per task). However, I am not exactly clear how this dataset tests compositionality and how it relates to other work that also tests compositionality (e.g., CLEVR). I look forward to the rebuttal.

---

> ### Author Response · Authors · 2022-08-17
> **Response to reviewer QftC [Part 1]**
>
> We thank the reviewer for the detailed feedback on the paper. Your feedback helped us improve the presentation of our work. We provide detailed responses to your concerns.
>
> [**A vague definition of compositionality**] We acknowledge the lack of a proper definition for compositionality or an explanation of its use in the benchmark and we thank the reviewer for the insightful references. We enriched the dataset description section with more information on compositionality and the related work section with comparisons to the papers referenced in the review. As described in [1] “The meaning of a complex expression is determined by the
> meanings of its constituents and by its structure”. Although compositionality is defined in the context of linguistics, this definition can be generalized to other modalities and abstract concepts. Our work evaluates compositionality over the reasoning operations necessary to solve visual reasoning problems. More specifically, the curriculum and reverse curriculum settings evaluate whether models leverage compositionality while learning to solve new tasks.
>
> [1] Szabó, Zoltán Gendler. "Compositionality as supervenience." Linguistics and Philosophy (2000): 475-505.
>
> [**Sample efficiency can be tested on any dataset**] We acknowledge that it is always possible to downsample training sets of other visual reasoning datasets. Our benchmark does not innovate in that regard and prior work on visual reasoning explores sample efficiency in a similar manner [1]. However, we argue that it is important to establish a benchmark where creating more sample efficient models is the main goal to encourage more research in this direction. This contribution is accompanied by our newly proposed sample efficiency score that favors more sample efficient models. Furthermore, the variety of rules in the dataset allows for an in-depth analysis of each model’s strengths and weaknesses in terms of sample efficiency, i.e, to identify which tasks require more samples to reach a high performance. These insights are more useful for guiding the development of future models than the sample efficiency over the entire dataset.
>
> [1] Kim, Youngsung, et al. "Few-shot visual reasoning with meta-analogical contrastive learning." Advances in Neural Information Processing Systems 33 (2020): 16846-16856.
>
> [**limited training set size**] Although we provide 10000 problems for each task in the dataset, it is possible to generate an infinite number of problems using our implementation. We choose to evaluate the models on a range from 20 to 1000 samples because the benchmark aims to promote more sample efficient visual reasoning models. However, as we mentioned in the paper L224, we train 2 models on 10000 samples and they achieve a high level of performance; SSL pre-trained ResNet-50 achieves 93.1% and SSL pre-trained ViT-small achieves 81.6%. We also provide new joint training results of all models on 10,000 samples without SSL pre-training. We observed an increase in performance for all models showing that all tasks are learnable, but models have inefficient inductive biases that render them overly reliant on data quantity. ResNet-50 achieves the highest level of performance at 93.7% compared to 59.2% with 1,000 samples. We will incorporate these new results in the supplementary material. Ideally, the benchmark would require estimating the lowest number of samples required by a given model to reach 100% performance. However, the search would be computationally expensive and we decided to evaluate accuracy at fixed dataset sizes instead.
>
> | Model (10000 samples) | accuracy     | N tasks > 80% |
> | -------               | -----              | -----         |
> | ResNet-50         | 93.7         | 93         |
> | ViT-Small             | 58.7         | 37         |
> | SCL                   | 56.9         | 34         |
> | WReN                  | 64.5         | 43         |
> | SCL-ResNet 18     | 78.9         | 73         |

---

> > ### Author Response · Authors · 2022-08-17
> > **Response to reviewer QftC [Part 2]**
> >
> > [**Why join-ViT underperforms ind-ViT**] We thank you for addressing this detail in model performance. We found that you share the remark with reviewer hShR. Thus, we paste the answer below:
> > Differences in accuracy between the joint and individual rule learning for a given model can be due to: 1) a model’s capacity to learn multiple tasks without interference and 2) a model's capacity to learn tasks by efficiently sharing knowledge between them.
> > In our experiments, we observed opposite trends for ResNets and ViTs -- with  ResNets/ViTs performing better/worse in the joint/individual rule-learning setting compared to the individual/joint rule-learning setting. We could explain this tendency in ResNets by their capacity to share knowledge among tasks allowing them to reach higher performance in lower data regimes. On the other hand, we believe that the opposite tendency in ViT can be explained by 1- the underperformance of transformers in low data regimes and 2- the sparse sampling of problems in the joint rule-learning setting. Simple rules such as the elementary position rule or the elementary size rule are easy to solve due to the patch embeddings in transformers. They are solved easily in the individual training setting since batches of inputs are based on the same rule. However, rules are sparsely sampled in the joint-training setting in each batch. We believe the sparse sampling of rules hinders performance in ViTs, especially on simple tasks.
> >
> > [**Model description**] We provide a detailed description of model training in the supplementary material Section 4 and Figure 3. ViT-small and ResNet-50 are fed the 4 images separately. The models are followed by an MLP that reduces the dimensionality of each image representation to 128. Then, the odd one out is chosen as the vector with the highest distance from the other 3 vectors. We also describe architectures in a new figure in the supplementary material (figure 4).
> >
> > **Final remark**: In conclusion, we thank you again for reviewing our paper and for the helpful recommendations. We hope that our response and clarification have addressed your questions and concerns about the definition of compositionality and sample efficiency tests. If so, we would appreciate it if you could consider raising the score. Please feel invited to engage with us if you have more questions.

---

### Official Review · Reviewer_hShR · 2022-07-28
**A novel benchmark for evaluation on critical capabilities of visual reasoning model, as well as empirical results for reference to better understand current visual reasoning models**

**Rating:** 6
**Confidence:** 3
**Correctness:** Yes
**Clarity:** Clear as a whole

**Strengths:**

1. Compositional generalization and sample efficiency are both significant topics in visual reasoning area. The paper focuses on the two issues and explores how performances on this benchmark are affected by different factors, e.g. architecture, pre-training, rule embedding.
2. The benchmark covers a wide range of tasks, most of which are clear and highly related to abstract reasoning, as well as multiple settings convenient for testing various levels of transferring or reasoning.
3. Incorporation of human performance helps construct a quantitative bar for reference, which further highlights the weakness and headroom for current visual reasoning models.

**Weaknesses:**

1. The motivation of rule embedding is not explained clearly, making this design not convincing enough.
2. Though huge efforts are made in task integration and experiments, this work lacks in-depth analysis about different architectures and intuitive explanation for discrepancies observed in results
3. The implementation is not informative enough, attaching relevant schematics or illustrations of models would be better.

**Additional Feedback:**

1. I am curious about why CNN-based models show different tendancy compared with Transformer-based models, e.g. between joint and individual rule embedding?
2. Similarly, the reason why CNN-based models outperform Transformer-based models seems to be the scale of ViT. On the other hand, ViT-Small is not commonly used in current visual reasoning tasks. So the selection of this baseline may not be quite reasonable.

**Documentation:**

Yes

**Relation To Prior Work:**

Yes

**Summary And Contributions:**

This paper introduces an benchmark for visual reasoning, or more specifically, the capabilities about abstraction of attributes and relations. It incorporates a compositionality prior to facilitate scene generation. It mainly explores how current models do in terms of generalization and sample efficiency. It conducts empirical studies on some baseline models, which compare the performances under different settings and indicate that current deep learning visual models still lag behind human performance a lot.

---

> ### Author Response · Authors · 2022-08-17
> **Response to reviewer hShR [Part 1]**
>
> Thank you for the effort you have invested in reviewing our work. Your perspective and critical feedback have helped us to extend and refine the paper. We provide a detailed response to your questions and comments in what follows.
>
> [**rule embedding**] This is an important question that we touched on in section 3 paragraph “Joint vs. Individual rule learning” L175. We will include an example in section 3 to clarify this choice.
>
> *... To illustrate this problem, let’s take the elementary size rule as an example. In this rule, each image contains one object. Due to the random sampling of object attributes, it is possible for one image to be considered an outlier with respect to the color rule (The attributes in the 4 images are i-small/green, ii-large/green, iii-small/green, iv-small/blue). Without specifying that the rule to solve is a size rule, the model could incorrectly choose the fourth image because it is an outlier with respect to the color rule. Thus, models trained on several tasks could easily confound rules …*
>
> [**lack of in-depth analysis**] We acknowledge the lack of in-depth analysis of the results in the paper. Although the paper’s primary focus is presenting a novel visual reasoning benchmark and baselines, it is important to suggest interpretations of the provided results. We analyzed task difficulty in Figure 6 and added a new analysis of pairwise differences between models over the experimental settings, pre-training, and architectures in the supplementary material’s Figure 9. We also try to explain model decisions using attribution methods in supplementary material’s Figure 10. However, we find that these methods are not useful for explaining decisions of neural network based models on CVR problem samples.
>
> [**schematic illustrations**] We have included model schematics in the supplementary material Figure 4.

---

> > ### Author Response · Authors · 2022-08-21
> > **Response to reviewer hShR [Part 2]**
> >
> > Responses on Additional feedback:
> >
> > [**CNNs vs. Transformers**]  Differences in accuracy between the joint and individual rule learning for a given model can be due to: 1) a model’s capacity to learn multiple tasks without interference and 2) a model's capacity to learn tasks by efficiently sharing knowledge between them.
> > In our experiments, we observed opposite trends for ResNets and ViTs -- with  ResNets/ViTs performing better/worse in the joint/individual rule-learning setting compared to the individual/joint rule-learning setting. We could explain this tendency in ResNets by their capacity to share knowledge among tasks, allowing them to perform higher in lower data regimes. On the other hand, we believe that the opposite tendency in ViT can be explained by 1- the underperformance of transformers in low data regimes and 2- the sparse sampling of problems in the joint rule-learning setting. Simple rules such as the elementary position rule or the elementary size rule are easy to solve due to the patch embeddings in transformers. They are solved easily in the individual training setting since batches of inputs are based on the same rule. However, rules are sparsely sampled in the joint-training setting in each batch. We believe the sparse sampling of rules hinders performance in ViTs, especially on simple tasks.
> >
> > [**Selection of ViT-small as a baseline**] We believe that ViTs have been used successfully in visual reasoning tasks at different scales [1,2,3]. Some small ViTs are competitive with SOTA on image recognition benchmarks [4]. Furthermore, it is generally better to use small architectures when training models on datasets with simple statistics (such as CVR; mostly white backgrounds) since large models overfit these datasets quickly through memorization. ResNet-50 and ViT-small were chosen because they have a similar number of parameters. They are carefully trained to avoid overfitting but they still achieve 100% training accuracy at the end of training. Thus, we can imagine that a larger model would overfit the dataset more easily, even with heavy regularization. However, if a large model is pre-trained on other visual tasks (like image annotation), it can achieve high performance on CVR, as shown in new results where ViT-Base is finetuned in the joint rule learning setting.
> >
> > | Model  | 20 | 50 | 100 | 200 | 500 | 1000 | SES | AUC |
> > | --- | --- | --- | --- | --- | --- | --- | --- | --- |
> > | resnet 50 (Imagenet) | 32.0 / 2 | 35.1 / 5 | 39.0 / 9 | 43.8 / 13 | 57.7 / 48 | 69.5 / 48 | 43.4 | 46.2 |
> > | vit small (Imagenet) | 27.9 / 2 | 28.2 / 1 | 28.6 / 2 | 30.0 / 0 | 35.6 / 5 | 47.2 / 24 | 31.7 | 32.9 |
> > | vit base (clip) | 31.1 / 1 | 37.4 / 7 | 43.9 / 14 | 56.0 / 30 | 68.9 / 48 | 78.8 / 62 | 48.9 | 52.7 |
> > | resnet 50 (clip) | 28.7 / 0 | 32.0 / 2 | 40.8 / 11 | 46.9 / 18 | 59.7 / 40 | 74.4 / 53 | 43.7 | 47.1 |
> >
> > **Final remark**: To conclude, we thank you again for the effort you have invested in reviewing our work. We hope that we have addressed your concerns properly. We are eager to discuss the submission further. If you agree that our revisions address your concerns about clarity and analysis, we would appreciate it if you could consider raising the score.
> >
> > [1] Messina, Nicola, et al. "Recurrent Vision Transformer for Solving Visual Reasoning Problems." International Conference on Image Analysis and Processing. Springer, Cham, 2022.
> >
> > [2] Li, Liunian Harold, et al. "Visualbert: A simple and performant baseline for vision and language." arXiv preprint arXiv:1908.03557 (2019).
> >
> > [3] Radford, Alec, et al. "Learning transferable visual models from natural language supervision." International Conference on Machine Learning. PMLR, 2021.
> >
> > [4] Yuan, Kun, et al. "Incorporating convolution designs into visual transformers." Proceedings of the IEEE/CVF International Conference on Computer Vision. 2021.

---

### Author Response · Authors · 2022-08-17
**General Comment**

We thank all the reviewers for their time and effort in reviewing our work. We are pleased to see that all reviewers recognize the significance of our topic and the value of the benchmark we propose. The reviewers agree on the importance of evaluating sample efficiency and compositionality in neural networks, the novelty of the task creation process, and the diversity of tasks in the benchmarks. Furthermore, they found our evaluation of SOTA models and humans thorough and insightful (reviewer hShR, QftC, FKHa, ehHC). They also agree on the benchmark's flexibility and potential for development.

Although the reviewers appreciate the premise and ideas of this work, they expressed concerns regarding the presentation and proposed several improvements to the methodology. The primary concern was a lack of clarity and details in the paper, especially in the description of the task generation process and the lack of analysis to explain the experimental results of the paper. We are grateful for the reviewers’ numerous questions that have helped us identify where our presentation can be further improved.

We have revised the paper to clarify the methodology and enriched the related work section. New figures were added to the manuscript and the supplementary material. We also ran experiments suggested by the referees to strengthen our initial results; training models with the more random seeds to compute the standard deviation of accuracy, training models on a higher data regime (10,000 samples), finetuning ImageNet pre-trained models and CLIP visual encoders on CVR, training ViT with a different optimization scheme, model explainability, and analysis of pairwise differences between models. We hope that the new results and updates to the manuscript will clarify our work and convince the reviewers of its correctness.

---

### Meta-Review · Area_Chair_ttsp · 2022-09-10

**Recommendation:** Accept
**Confidence:** 4

**Metareview:**

Most reviewers appreciate the importance of the topic (compositional visual reasoning), the flexibility of the benchmark (easily extended to incorporate different tasks), the thorough comparison, insightful experiments. However, the reviewers also raise questions about the clarity of the presentation, the limited training set size. AC read reviews and response from the authors and found that the concerns are addressed fairly well. AC recommends to accept the submission as a poster.

---

### Decision · Program_Chairs · 2022-09-16

Accept